DESY-24-201

# From the EFT to the UV: the complete dimension-six SMEFT one-loop dictionary

**Guilherme Guedes,**[a] **Pablo Olgoso**[b,c]

[a]*Deutsches Elektronen-Synchrotron DESY, Notkestr. 85, 22607 Hamburg, Germany*

[b]*Dipartimento di Fisica e Astronomia, Università di Padova, Via F. Marzolo 8, 35131 Padova, Italy*

[c]*Istituto Nazionale di Fisica Nucleare, Sezione di Padova, Via F. Marzolo 8, 35131 Padova, Italy*

*E-mail:* guilherme.guedes@desy.de, pablo.olgosoruiz@unipd.it

ABSTRACT: Effective field theories (EFTs) provide an excellent framework for the search of heavy physics beyond the Standard Model, using the so-called bottom-up and top-down approaches. However, the vastness of possible UV scenarios makes the complete connection between the two approaches a difficult challenge at the loop-level. UV/IR dictionaries fill precisely this gap, efficiently connecting the EFT with the UV. In this work we present the complete one-loop dictionary for the Standard Model EFT at dimension six for completions with an arbitrary number of heavy fermions and scalars. Our results (as well as several new functionalities) are added to the previously partial package SOLD, introduced in [1]. In this new form, SOLD is prepared to serve as an important guiding tool for systematic and complete phenomenological studies. To illustrate this, we use the package to explore possible explanations for the tension on the measurement of $\mathcal{B}(B \to K\overline{\nu}\nu)$.

## 1 From the EFT to the UV (and back)

The tremendous effort in the particle physics experimental programme of the past few years, in particular at the Large Hadron Collider (LHC), and the lack of an observation of a significant deviation from the Standard Model (SM) predictions, has given strength to the argument that new physics should be hiding behind a mass gap in regards to the electroweak scale. In the scenario of heavy physics beyond the SM (BSM), the use of Effective Field Theories (EFT), in particular of the Standard Model EFT (SMEFT) [2–5], proves to be an ideal framework.

Calculations in the SMEFT can in general be split into two independent steps. With the bottom-up approach, deviations from the SM are parameterized in terms of the Wilson Coefficients (WCs) of an EFT, with limited theoretical bias. This provides a model-independent interpretation of experimental data via global fits. Subsequently, the top-down approach translates these coefficients into the parameters of some specific candidate model, through the procedure known as matching. Thus, these two complementary perspectives allow us to connect our UV theories to experimental data.

These approaches have experienced an outstanding development in the past few years. In particular, the automation of many steps of the calculation [1, 6–12] has made it possible to almost streamline the process of comparing the predictions of a model with experimental data. However, the large number of possible models not only makes this comparison

cumbersome, losing in practice the efficiency of EFTs, but also blurs the interpretation of experimental data in terms of concrete models. Without any guidance, there is an effective gap between the two approaches.

Fortunately, EFTs also provide us a rationale to order, using power counting and perturbation theory, the relevance of different WCs, allowing the classification of all the new physics models (and only those) which are observable up to some order in the EFT expansion. This information constitutes a UV/IR dictionary, that comprises the translation between the models that contribute at some order to a specific operator and their explicit contribution. That is, dictionaries ultimately connect the bottom-up and top-down approaches as they answer the questions: *What are all models that can generate a specific operator (or a set of operators)? What are the low-energy consequences, through the matching conditions onto the SMEFT, of a specific UV model?*

The leading, complete tree-level dictionary for SMEFT at dimension six has been computed in Ref. [13], building on the work from Refs. [14–17]. However, given the increasing precision arising in experimental physics, it is important to also consider loop-level effects. This is mainly prompted by the fact that several observables receive their leading contributions at loop-level (for instance the anomalous magnetic moment of the muon [18–21]), but also from a theoretical perspective, as the mixing between tree- and loop-generated operators through renormalization has been extensively studied in the literature [22–26]. Furthermore, models with only quadratic couplings – which we define as couplings involving at least two BSM fields – can only contribute to the SMEFT at loop-level. These couplings are harder to probe and arise in models with a $Z_2$ symmetry, under which the BSM fields are odd, which can be found in models with a Dark Matter candidate [27, 28].

The construction of a complete one-loop SMEFT dictionary has been an ongoing effort in the literature – see Refs. [27, 29–33] for partial results. One of the main challenges is the fact that the number of relevant UV models at loop-level is in principle unbounded. For this reason, our first iteration in the construction of the one-loop dictionary resulted in the `Mathematica` package `SOLD` [1], which included only operators of the SMEFT with field-strength tensors (those whose leading contribution from renormalizable weakly-coupled UV theories is necessarily at the loop-level). In this work, we extend this work to now include all operators of the SMEFT at dimension six. A user of this dictionary can now therefore obtain complete answers to the questions stated above: for a particular UV model, what is the value of all SMEFT WCs? For a particular operator, what are all scalar and fermion multi-field extensions which can generate a non-zero WC?

While the first question can already be addressed by current matching softwares [8–10], `SOLD` provides an easier to use framework, as the user only needs to provide the SM gauge representation of the fields in the UV model (as opposed to having to implement a full Lagrangian). The second question is a completely new challenge that only `SOLD` addresses, allowing for much more systematic and complete phenomenological works. In regards to the previous version we have also included several more functions which allow for a schematic (and therefore much faster) construction of the low-energy picture of a specific UV model.

The general procedure and the new functions and challenges in this upgrade of `SOLD` are presented in section 2. In section 3, we show how `SOLD` can be used to easily draw the

low-energy picture of a specific UV scenario; while other matching tools are better equipped to fully match a specific model to the SMEFT, SOLD is useful when one is only interested in a restricted set of operators, when one needs to probe several models and does not want to produce different input models or when a schematic view is enough (details on this schematic view are given in this section). To promote the use of SOLD in a phenomenological study, in section 4, starting from an apparent tension in the experimental measurement of $\mathcal{B}(B \to K\overline{\nu}\nu)$ [34], we map extensions of the SM that could explain it, up to three-field extensions. While the literature has rightfully focused on tree-level solutions [35–38] (since the necessary WC is relatively large), we systematically evaluate whether a loop-level solution is also possible. This is important not only from the perspective of being complete but also because future experimental updates of this tension might lower the central value of the relevant WCs. This section should not be understood as a full study of a particularly motivated model, but rather as exemplifying how to use dictionaries as guiding principles in phenomenological studies. Finally we provide our conclusions and outlook in section 5.

## 2   Computing the one-loop dictionary

In Ref. [1] we introduced the first iteration of the SMEFT one-loop dictionary, which only included the SMEFT operators which cannot receive tree-level contributions under the assumption of a weakly-coupled renormalizable UV theory; in the Warsaw basis [3] this amounts to all operators that include a field-strength tensor [13]. Here we complete the effort to include all SMEFT operators, *i.e.* including the operators which can in principle also receive tree-level contributions.

A significant part of the details regarding the matching strategy employed in the development of the one-loop dictionary has already been introduced in Ref. [1]. For completeness, we briefly review it here, focusing on the new challenges posed by considering the full SMEFT Warsaw basis.

### 2.1   Matching procedure

We perform the matching to the SMEFT at dimension six following the diagrammatic off-shell approach; this amounts to computing one-light-particle-irreducible (1lPI) diagrams which are matched onto a SMEFT Green basis, which can then be reduced to a minimal physical basis (the Warsaw basis in our case) through field redefinitions. The relevant operators in the Warsaw basis and the redundant and evanescent operators which complete the Green's basis follow the same notation as introduced in Appendix D of the MatchmakerEFT manuscript [9].

As our goal is to create a complete dictionary which considers scalar and fermionic multi-field UV extensions of the SM[1], our starting point is a general theory, where particles are arranged in multiplets according to their spin, $\Psi_a$ for fermions and $\Phi_b$ for scalars; the indices $a$, $b$ runs over light and heavy particles in the multiplet. Since the remaining

---

[1]We do not include heavy vectors at this point, as their one-loop matching needs a complete model with the spontaneous symmetry breaking details which generate its mass. See also Ref. [39] for more details on one-loop matching of heavy vectors.

quantum numbers of the heavy fields are not specified, the gauge contraction between the fields is left undefined, characterized by a general Clebsch-Gordan (CG) tensor which will be calculated later in the computation. This generic renormalizable theory is described by the Lagrangian:

$$\begin{aligned}
\mathcal{L}_{\mathrm{UV}} =& \delta_{\Psi_a} \bar{\Psi}_a \Big[ \mathrm{i} \slashed{D} - M_{\Psi_a} \Big] \Psi_a + \delta_{\Phi_a} \Big[ |D_\mu \Phi_a|^2 - M_{\Phi_a}^2 |\Phi_a|^2 \Big] \\
&+ \sum_{\chi=L,R} \Big[ Y^\chi_{abc} \overline{\Psi}_a P_\chi \Psi_b \Phi_c + \widetilde{Y}^\chi_{abc} \overline{\Psi}_a P_\chi \Psi_b \Phi_c^\dagger \\
&\qquad\qquad + X^\chi_{abc} \overline{\Psi^c}_a P_\chi \Psi_b \Phi_c + \widetilde{X}^\chi_{abc} \overline{\Psi^c}_a P_\chi \Psi_b \Phi_c^\dagger + \mathrm{h.c.} \Big] \\
&+ \Big[ \kappa_{abc} \Phi_a \Phi_b \Phi_c + \kappa'_{abc} \Phi_a \Phi_b \Phi_c^\dagger + \lambda_{abcd} \Phi_a \Phi_b \Phi_c \Phi_d \\
&\quad + \lambda'_{abcd} \Phi_a \Phi_b \Phi_c \Phi_d^\dagger + \lambda''_{abcd} \Phi_a \Phi_b \Phi_c^\dagger \Phi_d^\dagger + \mathrm{h.c.} \Big],
\end{aligned} \tag{2.1}$$

where $P_{L,R} = (1 \mp \gamma^5)/2$, $\Psi^c \equiv \mathcal{C} \overline{\Psi}^T$ with $\mathcal{C}$ the charge conjugation matrix, $\delta_{\Psi_a}$ is $\mathbf{1}$ $(1/2 \times \mathbf{1})$ for complex fermions (Majorana fermions, such that $\Psi_a^c = \Psi_a$) and $\delta_{\Phi_a}$ is $\mathbf{1}$ $(1/2 \times \mathbf{1})$ for complex scalars (real scalars, with $\Phi_a^\dagger = \Phi_a$) scalars. As mentioned before, all couplings should be understood as including a CG tensor encoding the information of how the fields will be contracted. Only when the QNs of the heavy fields are specified will these CGs be computed. The underlying assumptions considered are that heavy fermions are vector-like, with both chiralities transforming in the same way and that there is no tree-level mixing between particles. Further information on the conventions used can be found in Ref. [1], including a discussion on the usage of naive dimensional regularization prescription to deal with $\gamma_5$.

With this theory, we perform the matching onto the Green's basis of the SMEFT diagrammatically. The matching of this generic theory to the SMEFT is stored in `SOLD` and does not need to be calculated again by the user. When projecting these results to a specific UV theory, the quantum numbers (QNs) of the heavy fields (an arbitrary number of them) have to finally be specified. Since Eq. (2.1) includes all possible renormalizable UV interactions, it is not necessary to create any sort of model file. With the QNs, `SOLD` makes use of the group theory package `GroupMath` [40] to calculate the CGs for the specified UV theory.

The matching results are given both in the redundant Green's basis and in the minimal Warsaw basis. The rules to perform the projection from the redundant to the minimal basis are included in `SOLD`.

The main difference in regards to the matching procedure laid out in the first version of `SOLD` [1] is that, given that we are considering *all* operators of the SMEFT, those that do not contain field-strength tensors can in principle receive tree-level contributions. As such, one-loop contributions to the kinetic terms of SM fields must be taken into account since upon canonical normalization, these loop contributions will change the tree-level generated WCs. We also include in this version of the package the option to perform the matching at tree-level – see section 2.5. To increase the performance, and because the set of UV models

that can give rise to tree-level WCs is finite [13], we compute the tree-level results in `SOLD` (finding full agreement with [13]) and store them in an accompanying file.

Finally, we also take into account the reduction of evanescent operators. Since we perform our loop calculations in $d$ dimensions, projecting the results to a physical basis can generate tree-level evanescent structures, that vanish in four dimensions but that are different from zero in $d$ dimensions, such that they produce finite contributions when inserted in EFT (divergent) loop amplitudes. The contribution from such structures is incorporated, as computed in [41], in the aforementioned rules for this projection.

## 2.2 Model classification

Addressing our second question – what are the SM extensions which can generate a non-zero WC – is much more challenging than in the tree-level case. The reason is that, at loop level, couplings that are quadratic in the heavy fields can contribute in the matching, generating topologies that do not fix the representations of the particles running in the loop, but only their product. The number of particular extensions is therefore technically unbounded. As a consequence, we report an answer to this question at two different levels: the first level collects all the different restrictions that a model can fulfill to contribute to a certain WC, and the second level lists the specific group representations that satisfy these restrictions.

In order to gather all this information, we make use of the intermediate result of the matching of the generic theory in section 2.1 onto the SMEFT. In this result, as explained in the previous section, all the gauge information is unspecified, in such a way that we can iterate over each diagram that can kinematically contribute to the WC of interest and extract the restrictions that the representations of the heavy particles in the loop need to fulfill for said diagram to be allowed by gauge symmetry. This extensive list is reduced to contain the minimum number of physically non-equivalent fields (that is, not related by conjugation) required to satisfy each restriction. This constitutes the first level result and defines all different "classes of models" that can generate a WC.

Regarding the second level, given a particular restriction, we check which specific representations can satisfy it using `GroupMath`. Notice that, by construction, it could happen that two different classes of models in the first level result contain the same specific model, that is, different restrictions can be respected by the same representations. Likewise, it could happen that for an $N$-field class of models, there exist some $(N-1)$-field solutions[2] (in terms of specific quantum numbers) as two fields having the same representations can satisfy the restriction.

This new version of the dictionary is more complete (as compared to [1]) not only because we answer the question for all operators of SMEFT, but also because we include several new functionalities. Among them, there is the option of listing all completions that can generate operators *without* small SM couplings (we do not exclude terms proportional to $SU(3)$ gauge coupling, $g_3$, and the top Yukawa, $y_t$), such that the result is proportional only to new physics interactions (and $g_3$ or $y_t$). All these results are given in electronic form via the latest version of the `SOLD` package. See section 2.5 for the details on the usage.

---

[2]In more generality, it could happen that in a class of $N$-fields, $M$ fields having the same QNs respects the restriction, meaning that we actually have a $(N-M+1)$-field extension.

## 2.3 Some general results

Operators which are composed only of field-strength tensors are generated by fields charged under the corresponding gauge symmetry. Since in the SMEFT the gauge group is considered unbroken, this results in the fact that only a single particle can run in the loop, making it possible to obtain general results for the WC of these operators, depending only on the spin and representation of the heavy particle. In Ref. [1], SOLD was used to rederive diagrammatically the general matching conditions of operators of the form $\mathcal{O}_X = X_\nu^\mu X_\rho^\nu X_\mu^\rho$ and $\mathcal{O}_{\tilde{X}} = X_\nu^\mu X_\rho^\nu \tilde{X}_\mu^\rho$, where $X_\nu^\mu$ corresponds the field-strength tensor associated to $SU(2)$ and $SU(3)$ gauge groups – in Ref. [42] these results had been obtained using functional methods. Similarly, results for the redundant operators $\mathcal{R}_{2X} = -1/2(D_\mu X^{\mu\nu})(D^\rho X_{\rho\nu})$, where now $X$ includes also the field-strength tensors associated with $U(1)$ gauge group, had also been derived in Ref. [42]. Once again we use SOLD to obtain these results diagrammatically, resulting in

$$\alpha_{2X} = \frac{1}{(4\pi)^2} \sum_R \frac{4c_R \, g^2}{15 M_R^2} \mu(R), \quad c_R = \begin{cases} 1, & \text{Dirac fermions} \\ \frac{1}{2}, & \text{Majorana fermions} \\ \frac{1}{8}, & \text{complex scalars} \\ \frac{1}{16}, & \text{real scalars} \end{cases}, \quad (2.2)$$

with $\text{Tr}(T_R^A T_R^B) = \mu(R)\delta^{AB}$, where $T_R$ are the generators of the group in $R$'s representation and $R$ iterates over the heavy fields in the model; $g$ is the gauge coupling constant associated to $X$. Ref. [42] further shows results for heavy vectors.

These are redundant operators with respect to the Warsaw basis. As such, their effect will be projected into this minimal basis [43] and will propagate to almost all operators without field-strength tensors (with the exceptions being $C_{Hud}$, $C_{quqd}^{(1,8)}$, $C_{\ell equ}^{(1,3)}$ and $C_{\ell edq}^{(1,3)}$) depending on the charges of the fields contained in the operators.

Besides the general relevance of performing this cross-check following an independent method, the fact that the matching of the operators $\mathcal{R}_{2X}$ propagates to several operators in the Warsaw basis, means that when using a one-loop dictionary to construct the UV theories potentially behind one of these operators, the result would always include any heavy particle charged under the SM gauge groups. This led us to also consider the possibility of constructing the UV theories that can generate a WC in the limit of vanishing SM couplings (in this case the gauge couplings are the relevant SM interactions). We explore in more detail these new functionalities in section 2.5.

## 2.4 Installing SOLD

The Mathematica package SOLD is publicly available in the Gitlab repository: https://gitlab.com/jsantiago_ugr/sold. In order to install SOLD, the user needs to install GroupMath [40] as well[3]. To install SOLD the following command is enough:

```
In[1]:= Import["https://gitlab.com/
        jsantiago_ugr/sold/-/raw/main/install.m"]
```

---

[3]To use the function in SOLD which performs the full one-loop matching using MatchmakerEFT, one would also need MatchmakerEFT to be installed.

Alternatively, the user might prefer to manually download the package from the repository and place it in the `Applications` folder of `Mathematica`'s base directory.

Afterwards, loading `SOLD` proceeds as with any other `Mathematica` package

In[2]:= << SOLD`

with an output shown in Fig. 1. Whenever `SOLD` is loaded, a comparison with the current version in the public repository is performed. If the versions do not match (that is, there has been an update in `SOLD`) the user is prompted to install the newer version.

In[1]:= **<< SOLD`**

SMEFT One Loop Dictionary loaded
Version: 2.0.1
Authors: Guilherme Guedes, Pablo Olgoso, José Santiago
Webpage: https://gitlab.com/jsantiago_ugr/sold
Reference: – SciPost Phys. 15 (2023) 143 *arXiv:2303.16965*
           – *arXiv:2412.14253*

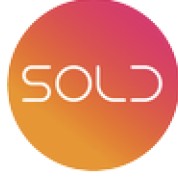

**Figure 1**: Loading `SOLD`.

## 2.5   List of new functions

Here we list only the new functions introduced in the new version of the `SOLD` package. Unchanged functions can be found in Ref. [1], or with the usual help command in `Mathematica` to obtain more information on all functions of the package. An updated version of the manual can be found in `SOLD`'s installation directory.

- `Match2GreenTree[coefficient, extension]`. Computes the tree-level contribution to a particular WC `coefficient` in the Green's basis (with notation as defined in `MatchmakerEFT` [9]) generated by a model defined by `extension`, where `extension` is a list of replacement rules specifying the information of the heavy particle content in our usual convention, i.e., with a tag (that must begin with an `S` or `F` depending on whether the heavy particle is a scalar or a fermion respectively and be followed by an identifying letter) and a list of its quantum numbers under the SM gauge group. Non-equivalent SU(3) representations of the same dimension, including conjugates, must be indicated with their Dynkin indices. We remind the reader that this notation for `coefficient` and `extension` holds for all functions.

- `Match2WarsawTree[coefficient,extension]`. Computes the tree-level contribution to `coefficient` in the Warsaw basis.

- `MatchSchematic[coefficient, extension,<listcouplings>]`. Returns a schematic result for the matching of `extension` into `coefficient` in the Warsaw basis. This is done by constructing the diagrams (with the couplings in `listcouplings`) which

contain the appropriate kinematic structure contributing to `coefficient`, but without computing them. The argument `listofcouplings` is optional; by default it will consider all possible couplings, which are themselves calculated through the function `CreateLag`. The same considerations apply to `MatchSchematicGreen` for coefficients in the Green's basis.

- `ListModelsWarsawTree[coefficient,<noSMcouplings>]`. Gives the list of heavy fermions and scalars that can generate `coefficient` at tree level. If the optional argument `noSMcouplings` is set to True, it only considers results without SM couplings (with the exception of the top yukawa and the strong gauge coupling) – it is set to False by default and in that case considers all possible couplings[4]. The same considerations apply to `ListModelsGreenTree`.

- `OpGeneratedQ[operator,model,<noSMcouplings>]`. Returns `True` if `model` is included in `ListModelsWarsaw[operator,noSMcouplings]`, i.e., it can possibly generate a contribution to `operator` in the Warsaw basis. `noSMcouplings` is an optional argument which, if set to `True`, considers results without SM couplings (with the exception of the top yukawa and the strong gauge coupling).

- `ListOperators[model,<noSMcouplings>]`. Given a `model`, prints the list of Warsaw coefficients that can be generated at tree-level and one-loop. `noSMcouplings` is an optional argument which, if set to `True`, considers results without SM couplings (with the exception of the top yukawa and the strong gauge coupling).

In addition, the following remarks should be taken into account with respect to the pre-existing functions:

- `Match2Warsaw[coefficient, extension]`. Since version 2.0.0, `coefficient` can be any dimension six SMEFT operator. As such, the results include (possibly) tree-level contributions. The tag `onelooporder` is used to separate the one-loop order contribution. The same considerations apply to `Match2Green`.

- `ListModelsWarsaw[coefficient,<noSMcouplings>]`. In version 2.0.0, `coefficient` can be any operator in the Warsaw basis. If the optional argument `noSMcouplings` is set to `True`, it returns the list of models that can give one-loop contributions to `coefficient` without Standard Model couplings (with the exception of the top yukawa and the strong gauge coupling). It is set to `False` by default. The same considerations apply to `ListModelsGreen`.

- `ListValidQNs[listrestrictions,<MaxDimSU3>,<MaxDimSU2>]`. When given any of the optional arguments `MaxDimSU3, MaxDimSU2` between curly brackets, returns results including representations *only* of the dimensions in brackets.

---

[4]Upon community feedback, we have also added an extra option for `noSMcouplings` which is `Gauge`, where the result is only considered with UV couplings, the top yukawa and all SM gauge couplings. This is to avoid light yukawa suppressed contributions.

Finally, we have added a parallelized version for the following functions, which greatly reduces their execution time:

- `Match2WarsawPar[coefficient, extension]`.

- `Match2GreenPar[coefficient, extension]`.

- `MatchSchematicPar[coefficient, extension, <listcouplings>]`.

- `MatchSchematicGreenPar[coefficient, extension, <listcouplings>]`.

## 3    The low-energy picture of the UV with `SOLD`

The purpose of this section is to give a practical example of how the new functions in `SOLD` were envisaged to give the user a quick picture of the phenomenology of a model, allowing to readily connect between top-down and bottom-up perspectives.

Let us consider a scenario in which some anomalous observation can be explained by a sizable new physics contribution to the operator $\mathcal{O}_{\ell q}^{(1)}$ (as we will actually explore in detail in the next section). Using the `ListModelsWarsawTree` function, we can check the list of models that generate it a tree level:

In[3]:= `ListModelsWarsawTree[alphaOlq1[i,j,k,l]]`

$$
\begin{pmatrix}
\texttt{FieldContent} & SU(3) \otimes SU(2) & U(1) \\
\{\phi 1\} & \{\phi 1 \to 3 \otimes 1\} & \{\texttt{Y}_{\phi 1} \to -\frac{1}{3}\} \\
\{\phi 1\} & \{\phi 1 \to 3 \otimes 3\} & \{\texttt{Y}_{\phi 1} \to -\frac{1}{3}\}
\end{pmatrix}
\tag{3.1}
$$

In a slight abuse of notation, we use $R_{SU(3)} \otimes R_{SU(2)}$ to denote the representations, $R$, of the fields under the $SU(3)$ and $SU(2)$ gauge groups of the SM, which can also be understood as $(R_{SU(3)}, R_{SU(2)})$.

As it is well known [13], we can see that the $S_1 \sim (3, 1, -1/3)$ leptoquark can contribute to this operator at tree level. We can obtain the explicit result using the following function:

In[4]:= `Match2WarsawTree[alphaOlq1[i,j,k,l], {Sa->{3,1,-1/3}}]`

Out[4]= $\dfrac{\texttt{L1[Sabar, lL, qL][j, l] L1bar[Sabar, lL, qL][i, k]}}{4 \ \texttt{MSa}^2}$

The coupling notation is the same as introduced in Ref. [1]. Couplings are defined by the letter $L$ followed by a number (these numbers distinguish between different gauge contractions between the same fields). The first argument corresponds to the particles comprising the operator to which the coupling corresponds to and the second argument corresponds to the flavor indices of the operator. To explicitly see the definition of the couplings, the function `CreateLag[{Sa->{3,1,-1/3}}]` outputs the Lagrangian of the model.

Using `SOLD`, one can also check the list of all possible models that can generate this operator at one loop:

In[5]:= `ListModelsWarsaw[alphaOlq1[i,j,k,l]]`

$$
\begin{pmatrix}
\texttt{FieldContent} & \mathrm{SU}(3)\otimes\mathrm{SU}(2) & \mathrm{U}(1) \\
\{\phi1\} & \{\phi1 \to \overline{3}\otimes 1\} & \{\mathrm{Y}_{\phi1} \to \tfrac{1}{3}\} \\
\{\phi1\} & \{\phi1 \to \overline{3}\otimes 3\} & \{\mathrm{Y}_{\phi1} \to \tfrac{1}{3}\} \\
\{\phi1\} & \{\phi1 \to 1\otimes 1\} & \{\mathrm{Y}_{\phi1} \to 1\} \\
\{\phi1\} & \{\phi1 \to 1\otimes 2\} & \{\mathrm{Y}_{\phi1} \to -\tfrac{1}{2}\} \\
 & \ldots & \\
\{\phi1,\psi1\} & \{\psi1\otimes\overline{\psi1} \supset 1\otimes 1 \,,\; \psi1\otimes\overline{\phi1} \supset \overline{3}\otimes 2\} & \{\mathrm{Y}_{\psi1} \to -\tfrac{1}{6} + \mathrm{Y}_{\phi1},\, \mathrm{Y}_{\psi1} \neq 0\} \\
 & \ldots &
\end{pmatrix}
$$
$$(3.2)$$

The first column gives information about the field content of the particular "class" of models, indicating how many scalars ($\phi\mathtt{i}$) and fermions ($\psi\mathtt{i}$) are present. The second column gives the list of the $\mathrm{SU}(3)\otimes\mathrm{SU}(2)$ restrictions that the fields have to satisfy to generate the particular operator (in our case, $\mathcal{O}_{\ell q}^{(1)}$). In some cases the representations are fixed (as in the first rows), whereas in others we have restrictions in the product of representations. The last column gives the same information for the hypercharge. The function `ListValidQNs` gives a list of specific quantum numbers that satisfy the restrictions given by `ListModels`, and `SOLDInputForm` can be used to transform its output into the appropriate input form for the matching functions. See section 2.5 and [1] for more details.

Using the new optional argument, we can actually see how this leptoquark generates $\mathcal{O}_{\ell q}^{(1)}$ even without SM couplings:

In[6]:= `ListModelsWarsaw[alphaOlq1[i,j,k,l],True]`

$$
\begin{pmatrix}
\texttt{FieldContent} & \mathrm{SU}(3)\otimes\mathrm{SU}(2) & \mathrm{U}(1) \\
\{\phi1\} & \{\phi1 \to \overline{3}\otimes 1\} & \{\mathrm{Y}_{\phi1} \to \tfrac{1}{3}\} \\
\{\phi1\} & \{\phi1 \to \overline{3}\otimes 3\} & \{\mathrm{Y}_{\phi1} \to \tfrac{1}{3}\} \\
\{\phi1\} & \{\phi1 \to 1\otimes 2\} & \{\mathrm{Y}_{\phi1} \to -\tfrac{1}{2}\} \\
 & \ldots &
\end{pmatrix}
$$
$$(3.3)$$

In order to assess the impact of such a completion in other observables, one needs to check which other operators are generated. We could quickly check, for instance, whether this model generates a contribution to the anomalous magnetic moment of the muon by means of the following command:

In[7]:= `OpGeneratedQ[alphaOeW[i,j], {Sa->{3,1,-1/3}}]`

Out[7]= `True`

This operator is never generated, however, in the limit of no SM couplings, since it is always proportional to the weak gauge coupling:

In[8]:= `OpGeneratedQ[alphaOeW[i,j], {Sa->{3,1,-1/3}},True]`

Out[8]= `False`

Using the function `MatchSchematic`, one can have a schematic idea of how the result for this coefficient could look like:

```
In[9]:= MatchSchematic[alphaOeW[i,j], {Sa->{3,1,-1/3}}]
```

$$\text{Out[9]= } \frac{\text{gw onelooporder L[eR, }\phi\text{, }\overline{\text{lL}}\text{]L[lL,qL, }\overline{\text{Sa}}\text{]L[Sa, }\overline{\text{lL}}\text{, }\overline{\text{qL}}\text{]}}{16 \text{ M}^2 \text{ } \pi^2}$$
$$+ \frac{\text{gw onelooporder L[eR, uR, }\overline{\text{Sa}}\text{]L[qL,}\phi\text{, }\overline{\text{uR}}\text{]L[Sa, }\overline{\text{lL}}\text{, }\overline{\text{qL}}\text{]}}{16 \text{ M}^2 \text{ } \pi^2}$$
$$+ \frac{\text{gw onelooporder xRP L[eR, uR, }\overline{\text{Sa}}\text{]L[qL,}\phi\text{, }\overline{\text{uR}}\text{]L[Sa, }\overline{\text{lL}}\text{, }\overline{\text{qL}}\text{]}}{16 \text{ M}^2 \text{ } \pi^2}$$

The numerical factors in this result are purely schematic (they are not actually computed); the $\frac{1}{16\pi^2}$ factors serve to state whether the (would-be) contribution corresponds to a tree-level or loop-level diagram. The couplings are given in a "schematic" form, indicating the fields comprising the associated operator. The parameter xRP is related to the reading point prescription used in the reduction of evanescent operators (see [1, 41] for details). This function admits an optional argument specifying the couplings (beyond the SM) that the user wants included in the result:

```
In[10]:= MatchSchematic[alphaOeW[i,j], {Sa->{3,1,-1/3}},{L[eR, ϕ, lL],L[lL,qL, Sa],
         L[Sa, lL, qL]}]
```

$$\text{Out[10]= } \frac{\text{gw onelooporder L[eR, }\phi\text{, }\overline{\text{lL}}\text{]L[lL,qL, }\overline{\text{Sa}}\text{]L[Sa, }\overline{\text{lL}}\text{, }\overline{\text{qL}}\text{]}}{16 \text{ M}^2 \text{ } \pi^2}$$

These couplings can be given schematically; the order of the fields inside the coupling and the writing of a coupling or its conjugate version is irrelevant.

Note that the actual result is given by the following function:

```
In[11]:= Match2WarsawPar[alphaOeW[i,j], {Sa->{3,1,-1/3}}]/.Log[__]->0
         /.onelooporder->1//NiceOutput
```

$$\frac{9g_2 \, (y_u)^{\dagger[fl2,fl1]} \bar{\lambda}_{\overline{Sa},lL,qL}^{[a,\,fl1]} \lambda_{\overline{Sa},eR,uR}^{[b,\,fl2]}}{256\pi^2 M_{Sa}^2} + \frac{g_2 y_e^{[flc,b]} \bar{\lambda}_{\overline{Sa},lL,qL}^{[a,\,fl1]} \lambda_{\overline{Sa},lL,qL}^{[flc,\,fl1]}}{128\pi^2 M_{Sa}^2}$$
$$+ \frac{3g_2 \xi_{rp} \, (y_u)^{\dagger[p,t]} \bar{\lambda}_{\overline{Sa},lL,qL}^{[a,\,t]} \lambda_{\overline{Sa},eR,uR}^{[b,\,p]}}{128\pi^2 M_{Sa}^2} - \frac{3g_2 \, (y_u)^{\dagger[p,t]} \bar{\lambda}_{\overline{Sa},lL,qL}^{[a,\,t]} \lambda_{\overline{Sa},eR,uR}^{[b,\,p]}}{128\pi^2 M_{Sa}^2}$$

Nevertheless, up to numerical factors and flavor structure, we can see how both results agree with each other.

Finally, one can access the complete list of operators generated by this model, both at tree and one-loop level, using the ListOperators function, that also includes an optional argument to consider results without SM couplings (again, including $g_3$ and $y_u$, the former being the strong gauge coupling and the latter being the up-quark Yukawa which includes the large top Yukawa). The output of this function for the $S_1$ leptoquark is given in Fig. 2.

## 4   Carving out the UV with SOLD: one-loop solutions to $\mathcal{B}(B \to K\bar{\nu}\nu)$

The measurement of a tension between experiment and SM prediction is always followed by an incredible effort to map the BSM scenarios that can be behind such an experimental

Out[2]=

| Name | Tree | Loop |
|------|------|------|
| $O_{3G}$ | | ✓ |
| $O_{3Gt}$ | | |
| $O_{3W}$ | | |
| $O_{3Wt}$ | | |
| $O_{HB}$ | | |
| $O_{HBt}$ | | |
| $O_{HG}$ | | ✓ |
| $O_{HGt}$ | | |
| $O_{HW}$ | | |
| $O_{HWB}$ | | |
| $O_{HWBt}$ | | |
| $O_{HWt}$ | | |
| $O_{dB}$ | | |
| $O_{dG}$ | | ✓ |
| $O_{dW}$ | | |
| $O_{eB}$ | | |
| $O_{eW}$ | | |
| $O_{uB}$ | | |
| $O_{uG}$ | | ✓ |
| $O_{uW}$ | | |

| Name | Tree | Loop |
|------|------|------|
| $O_{HBox}$ | | ✓ |
| $O_{HD}$ | | ✓ |
| $O_{lambda}$ | ✓ | ✓ |
| $O_{muH2}$ | | ✓ |
| $O_{dd}$ | | ✓ |
| $O_{dH}$ | | ✓ |
| $O_{ed}$ | | ✓ |
| $O_{ee}$ | | ✓ |
| $O_{eH}$ | | ✓ |
| $O_{eu}$ | ✓ | ✓ |
| $O_{Hd}$ | | ✓ |
| $O_{He}$ | | ✓ |
| $O_{Hl1}$ | | ✓ |
| $O_{Hl3}$ | | ✓ |
| $O_{Hq1}$ | | ✓ |
| $O_{Hq3}$ | | ✓ |
| $O_{Hu}$ | | ✓ |
| $O_{Hud}$ | | ✓ |
| $O_{lambdad}$ | | ✓ |
| $O_{lambdae}$ | | ✓ |

| Name | Tree | Loop |
|------|------|------|
| $O_{lambdau}$ | | ✓ |
| $O_{ld}$ | | ✓ |
| $O_{le}$ | | ✓ |
| $O_{ledq}$ | | ✓ |
| $O_{lequ1}$ | ✓ | ✓ |
| $O_{lequ3}$ | ✓ | ✓ |
| $O_{ll}$ | | ✓ |
| $O_{lq1}$ | ✓ | ✓ |
| $O_{lq3}$ | ✓ | ✓ |
| $O_{lu}$ | | ✓ |
| $O_{qd1}$ | | ✓ |
| $O_{qd8}$ | | ✓ |
| $O_{qe}$ | | ✓ |
| $O_{qq1}$ | ✓ | ✓ |
| $O_{qq3}$ | ✓ | ✓ |
| $O_{qu1}$ | | ✓ |
| $O_{qu8}$ | | ✓ |
| $O_{quqd1}$ | ✓ | ✓ |
| $O_{quqd8}$ | ✓ | ✓ |
| $O_{ud1}$ | ✓ | ✓ |
| $O_{ud8}$ | ✓ | ✓ |
| $O_{uH}$ | | ✓ |
| $O_{uu}$ | | ✓ |

**Figure 2**: The output of `ListOperators` for a leptoquark $S_a - > \{3, 1, -1/3\}$. The optional argument `True` shows results without considering SM couplings, except $g_3$ and $y_u$. The cells with the green "tick" indicate that the operator is generated at tree or loop level, correspondingly.

anomaly. In the same vein, a measurement which leads to a very stringent bound on a set of WCs, leads to an effort to understand which models are subject to such constraints. This systematic and complete description of all relevant UV scenarios is made possible by the use of dictionaries, Ref. [13] at tree-level and `SOLD` at loop-level.

Considering a specific observable – leading to a set of WCs – one wants to explain, `SOLD` can easily provide all the (single- and multi-field) scalar and fermionic extensions which generate those non-zero WCs, with the function `ListModelsWarsaw[c]`, where `c` is the relevant WC. Note that this does not perform any calculations as it has been hard-coded in the package; `SOLD` is acting as a database, or better yet, a dictionary. The output is the complete list of heavy-fields extensions which are defined by their content and the quantum numbers of these new particles.

After obtaining the list of models which *can* be behind a non-zero WC, one can then

scan over all of them to obtain the matching conditions of the desired WC. This scanning can be done very efficiently because `SOLD` does not require the input of any sort of Lagrangian, which would make it cumbersome to go over several dozens of models. Indeed, `SOLD` only needs the field content and quantum numbers of a particular SM extension to calculate matching conditions – exactly the output that one obtained before with the `ListModelsWarsaw[c]` function. `SOLD` then assumes that all gauge-invariant couplings of the UV theory are non-zero and matches the theory to the relevant WC following the strategy outlined in section 2.

To realize these procedure, let us consider the recent measurement of $\mathcal{B}(B \to K\overline{\nu}\nu)$ by Belle II [34] which found a result $2.9\,\sigma$ above the SM prediction,

$$R_{\nu\nu}^K = \frac{\mathcal{B}(B \to K\overline{\nu}\nu)}{\mathcal{B}(B \to K\overline{\nu}\nu)|_{\mathrm{SM}}} = 5.4 \pm 1.5\,. \tag{4.1}$$

There has been an effort in the literature to understand whether this possible anomaly could arise from heavy physics. In Refs. [35–38], the contributions from SMEFT operators were considered; given that the deviation appears to be relatively large, efforts to connect these non-zero WCs with UV scenarios have focused on tree-level completions. However, as will be explored henceforth, it is not clear that a loop-suppressed explanation is not possible.

In Ref. [44], a partial $\chi^2$-analysis was performed considering some $B$ decays (including $R_{\nu\nu}^K$ and $R_{D(*)}$) and the best fit point was found to be:

$$\begin{aligned}
\left[C_{\ell q}^{(1)}\right]_{2232} &= 6.5 \times 10^{-4} & \left[C_{\ell q}^{(1)}\right]_{3332} &= 5.57 \times 10^{-2} \\
\left[C_{\ell q}^{(3)}\right]_{3332} &= 4.75 \times 10^{-2} & \left[C_{\ell d}^{(1)}\right]_{3332} &= 1.87 \times 10^{-2}\,,
\end{aligned} \tag{4.2}$$

where we include only the central values and the high-energy scale was taken to be $\Lambda = 1\,\mathrm{TeV}$. The fit favors a sizable $\left[C_{\ell q}^{(3)}\right]_{3332}$ to explain $R_{D(*)}$ and $\left[C_{\ell d}^{(1)}\right]_{3332}$ with the approximate pattern $C_{\ell q}^{(1)} \sim C_{\ell q}^{(3)}$ to accommodate $R_{\nu\nu}^K$. See also Ref. [36] for a discussion of these patterns.

Neglecting heavy vector extensions, explaining these WCs at tree-level involves 3-fields. To generate $C_{\ell d}$ the only option is $\Pi_1 \sim (3, 2, 1/6)$; adding also $S_3 \sim (3, 3, 1/3)$ would generate $C_{\ell q}^{(1)} = 3C_{\ell q}^{(3)}$. Therefore, one would need a 3-field extension, adding also $S_1 \sim (3, 1, 1/3)$ to arrive at $C_{\ell q}^{(1)} \sim C_{\ell q}^{(3)}$.

While the large values of these WCs may point to this tree-level UV, a coefficient of order of magnitude $\sim \mathcal{O}(10^{-2})$ could in principle also arise from loop-level matching. In spite of the unavoidable loop suppression, the UV couplings of these models are usually much more weakly constrained than the tree-level models as the former can couple non-linearly to the SM, *i.e.*, UV couplings can be composed of two heavy particles and only one SM field (BSM models which only couple non-linearly cannot contribute at tree-level to the SMEFT). While it would still be difficult to explain the central values in this manner, being within 1- or 2-$\sigma$ regions of the fit could in principle be achievable given this bigger freedom on the UV couplings.

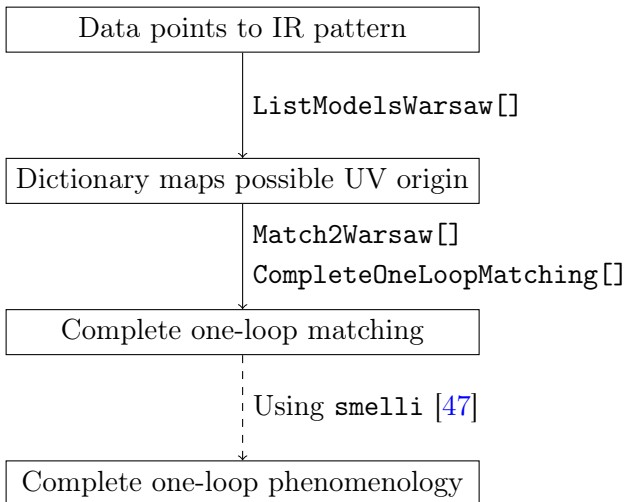

**Figure 3**: How `SOLD` can be used for phenomenological studies. For the last step, we use an in-house code to parse the one-loop matching results into a numerical input to `smelli`; because this feature is not implemented in `SOLD` we keep this arrow as dashed to represent that it is not yet automatized.

As such, let us proceed with an exploratory scan over the UV possibilities that can generate the pattern of Eq. (4.2), be it purely through loop solutions or with a mix of tree- and loop-level contributions. The procedure will follow the schematic illustration of Fig. 3: *i*) we use `SOLD` to construct the possible multi-field extensions behind one or more relevant operators; *ii*) still within `SOLD`, we compute the explicit matching contribution for some operators and also use `MatchmakerEFT` to compute the full one-loop matching conditions of promising models; *iii*) we connect the output of `MatchmakerEFT` with `smelli` [45–47] to assess whether the model is in agreement with other experimental constraints. Other fitting tools could be considered in this last step [11, 48–52]. There will be some tension in explaining the pattern Eq. (4.2) when comparing with other flavour bounds; this difficulty has been explored in more detail in Refs. [36, 44]. We will also comment on it throughout the section.

Let us start by considering the simplest solutions, models which consider only one heavy particle contributing at one-loop to these operators.

## 4.1 One-field extensions

Given that we aim to explain a relatively large numerical value for the WC, it is reasonable to make the simplifying assumption that the matching condition should not include any SM coupling apart from $y_t$ and $g_3$, the top Yukawa and the gauge coupling of $SU(3)$ respectively. It is not enough to look at the generation of $C_{\ell q}^{(1)}$ in the Green's basis since it can also receive contribution from evanescent redundant operators.

Let us then make use of the dictionary to obtain the possible one-field solutions, in the limit of vanishing small SM couplings, with the function

$$
\begin{pmatrix}
\text{Field Content} & SU(3)\otimes SU(2) & U(1) \\
\{\phi 1\} & \{\phi 1 \to \overline{\mathbf{3}}\otimes \mathbf{1}\} & \{Y_{\phi 1} \to \tfrac{1}{3}\} \\
\{\phi 1\} & \{\phi 1 \to \overline{\mathbf{3}}\otimes \mathbf{3}\} & \{Y_{\phi 1} \to \tfrac{1}{3}\} \\
\{\phi 1\} & \{\phi 1 \to \mathbf{1}\otimes \mathbf{2}\} & \{Y_{\phi 1} \to -\tfrac{1}{2}\} \\
\{\phi 1\} & \{\phi 1 \to \mathbf{3}\otimes \mathbf{2}\} & \{Y_{\phi 1} \to \tfrac{7}{6}\} \\
\{\psi 1\} & \{\psi 1 \to \mathbf{1}\otimes \mathbf{1}\} & \{Y_{\psi 1} \to 0\} \\
\{\psi 1\} & \{\psi 1 \to \mathbf{1}\otimes \mathbf{1}\} & \{Y_{\psi 1} \to 1\} \\
\{\psi 1\} & \{\psi 1 \to \mathbf{1}\otimes \mathbf{3}\} & \{Y_{\psi 1} \to 0\} \\
\{\psi 1\} & \{\psi 1 \to \mathbf{1}\otimes \mathbf{3}\} & \{Y_{\psi 1} \to 1\} \\
\end{pmatrix}
$$

**Figure 4**: The output from the function `ListModelsWarsaw[alphaOlq1[i, j, k, l],True]`. Here, we show only the one-field extensions which can potentially generate the WC $C^{(1)}_{\ell q}$. The first column identifies the spin of the field content, whereas the second and third columns the SM gauge representations of the heavy fields.

In[12]:= `ListModelsWarsaw[alphaOlq1[i, j, k, l], True]`

with the partial output (including only the one-field extensions) of Fig. 4. The first column defines the field content of the model (in these one-field extensions, it simply identifies if it is a scalar or fermion), the second and third columns define the representation under $SU(3)_C$ and $SU(2)_L$, and $U(1)_Y$, respectively.

We can further look for possible intersection with the output from `ListModelsWarsaw[alphaOlq1[i, j, k, l],True]` to find models which can also generate $\mathcal{C}_{\ell d}$. We are left with two possible extensions: $\Phi \sim (1, 2, 1/2)$ or $S_1 \sim (3, 1, -1/3)$. We will not consider the $S_1 \sim (3, 1, -1/3)$ extension as it generates $C^{(1)}_{\ell q} = -C^{(3)}_{\ell q}$ at tree-level [13]. With `Match2Warsaw` we can obtain the matching conditions for $\Phi \sim (1, 2, 1/2)$:

In[13]:= `Match2Warsaw[alphaOlq1[3, 3, 3, 2], {Sa -> {1, 2, 1/2}}]//`
`NiceOutput`

$$
\left[C^{(1)}_{\ell q}\right]_{3332} = \frac{\lambda_{\overline{\mathrm{lL}},\mathrm{eR},\mathrm{Sa}}^{[3,\,\mathrm{fl1}]}\,\bar{\lambda}_{\overline{\mathrm{lL}},\mathrm{eR},\mathrm{Sa}}^{[3,\,\mathrm{fl1}]}\,\lambda_{\overline{\mathrm{uR}},\mathrm{qL},\mathrm{Sa}}^{[\mathrm{fl2},\,2]}\,\bar{\lambda}_{\overline{\mathrm{uR}},\mathrm{qL},\mathrm{Sa}}^{[\mathrm{fl2},\,3]}}{128\pi^2 M^2_{\mathrm{Sa}}}
$$
$$
- \frac{\bar{\lambda}_{\overline{\mathrm{qL}},\mathrm{dR},\mathrm{Sa}}^{[2,\,\mathrm{fl2}]}\,\lambda_{\overline{\mathrm{lL}},\mathrm{eR},\mathrm{Sa}}^{[3,\,\mathrm{fl1}]}\,\bar{\lambda}_{\overline{\mathrm{lL}},\mathrm{eR},\mathrm{Sa}}^{[3,\,\mathrm{fl1}]}\,\lambda_{\overline{\mathrm{qL}},\mathrm{dR},\mathrm{Sa}}^{[3,\,\mathrm{fl2}]}}{128\pi^2 M^2_{\mathrm{Sa}}} + \cdots , \qquad (4.3)
$$

with $C^{(3)}_{\ell q} = C^{(1)}_{\ell q}$ and

In[14]:= `Match2Warsaw[alphaOld[3, 3, 3, 2], {Sa -> {1, 2, 1/2}}]//NiceOutput`

$$
[C_{\ell d}]_{3332} = \frac{\lambda_{\overline{\mathrm{lL}},\mathrm{eR},\mathrm{Sa}}^{[3,\,\mathrm{fl2}]}\,\bar{\lambda}_{\overline{\mathrm{lL}},\mathrm{eR},\mathrm{Sa}}^{[3,\,\mathrm{fl2}]}\,\lambda_{\overline{\mathrm{qL}},\mathrm{dR},\mathrm{Sa}}^{[\mathrm{fl1},\,2]}\,\bar{\lambda}_{\overline{\mathrm{qL}},\mathrm{dR},\mathrm{Sa}}^{[\mathrm{fl1},\,3]}}{64\pi^2 M^2_{\mathrm{Sa}}} + \cdots , \qquad (4.4)
$$

where $\cdots$ correspond terms proportional to SM couplings which we neglect. The coupling notation is the same that `SOLD` outputs; the subscripts of the couplings correspond to the fields comprising the corresponding operator and the superscript are the flavour indices. To better understand what the couplings actually look like in a Lagrangian, one can use

the function `CreateLag[model]` to see the couplings and gauge contractions that `SOLD` considers.

While this model in principle generates the relevant pattern between WCs, it seems difficult to generate a large numerical value for them, as implied by the $\chi^2$-fit in (4.2). Indeed, in this model all BSM couplings are linear (couplings involve only one BSM field and 2 or more SM particles), which are more constrained and lead to the tree-level generation of several operators. With this in mind, we do not find a reasonable parameter point that can lead to $C_{\ell q}^{(1,3)} = \mathcal{O}(10^{-2})$.

As such, the extra suppression we obtained (beyond the naive $1/(16\pi^2)$) makes it unfeasible to provide the necessary pattern with one field at one-loop level. To also consider quadratic couplings (those involving two BSM fields), which can in principle take larger numerical values, we are led to consider two-field extensions.

## 4.2 Two-field extensions

Considering extensions comprised of two heavy fields significantly increases the space of possible UV theories. Let us start by exploring models which generate $C_{\ell q}^{(1)} \approx C_{\ell q}^{(3)}$ at tree-level and $C_{\ell d}$ at loop-level. From the tree-level dictionary [13], there is only one model (excluding vectors) that results in $C_{\ell q}^{(1)}$ and $C_{\ell q}^{(3)}$ with the same sign, $S_3 \sim (3, 3, -1/3)$, which results in $C_{\ell q}^{(1)} = 3C_{\ell q}^{(3)}$. To explore how we can complement this model to generate $C_{\ell d}$ at loop-level, we can go over the output of `ListModelsWarsaw` and pick models that include $S_3 \sim (3, 3, -1/3)$. This can be done with the code:

```
In[15]:= listmodels=ListValidQNs[ListModelsWarsaw[alphaOld[i, j, k, l]
        , True][[ ;; ]], 3, 3];
        listmodelsinput=Flatten[Table[Table[(SOLDInputForm[#] & /@ ii2)
         , {ii2, ii1}], {ii1, listmodels}],1];
        Select[listmodelsinput, Length[#] === 2 &];
        models=Select[%,
        MatchQ[#, ({___, Sa_ -> {{1, 0}, 3, -1/3}, ___} |
           {___, Sa_ -> {{0, 1}, 3, 1/3}, ___}) /;
            StringContainsQ[ToString[Sa], "S"]] &];
```

where we are extracting the list of two-field extensions in `SOLD` input form (up to triplet representations) and selecting those that include $S_3 \sim (3, 3, -1/3)$ (or the conjugate). We now have to verify what are the actual matching conditions for these models; to this end, we run

```
In[16]:= Match2Warsaw[alphaOld[3, 3, 3, 2], #]&/@ models
```

which will output the WC for the relevant models. This command runs for approximately 6 minutes[5]; without specifying any model or creating any model file, the user can obtain the matching results for *all* two-field extensions which include $S_3$. This exercise exemplifies

---

[5]This benchmark was obtained with a 11th Gen Intel(R) Core(TM) i7-11850H @ 2.50GHz processor.

the main benefits of SOLD: it allows the calculation of specific WCs and it requires minimal user input.

We obtain nine possible models, including models comprised of two scalars or of one scalar and one fermion. Among them, the models with two scalars are not promising – the resulting WC includes only linear couplings to the SM (therefore more constrained couplings) and small numerical factors. Moving on to the scalar and fermion extensions, let us focus on the following two, $S_3 + Q_5$ with $Q_5 \sim (3, 2, -5/6)$ and $S_3 + T_2$ with $T_2 \sim (3, 3, 2/3)$:

In[17]:=  `Limit[Match2Warsaw[`
          `alphaOld[3, 3, 3, 2], {Sa -> {3, 3, -1/3},`
          `Fa -> {3, 2, -5/6}}], MFa -> MSa]/.onelooporder->1 // NiceOutput`

$$[C_{\ell d}]_{3332} = -\frac{3\lambda_{\bar{\phi},\mathrm{dR,Fa}}^{[2]}\bar{\lambda}_{\bar{\phi},\mathrm{dR,Fa}}^{[3]}\lambda_{\overline{\mathrm{Sa}},\mathrm{Fa,lL}}^{[3]}\bar{\lambda}_{\overline{\mathrm{Sa}},\mathrm{Fa,lL}}^{[3]}}{256\pi^2 M_{\mathrm{Sa}}^2} + \cdots, \qquad (4.5)$$

In[18]:=  `Limit[Match2Warsaw[`
          `alphaOld[3, 3, 3, 2], {Sa -> {3, 3, -1/3},`
          `Fa -> {3, 3, 2/3}}], MFa -> MSa]/.onelooporder->1 // NiceOutput`

$$[C_{\ell d}]_{3332} = \frac{3\lambda_{\mathrm{dR,Fa,Sa}}^{[2]}\bar{\lambda}_{\mathrm{dR,Fa,Sa}}^{[3]}\lambda_{\overline{\mathrm{Sa}},\mathrm{lL,qL}}^{[3,\,\mathrm{fl1}]}\bar{\lambda}_{\overline{\mathrm{Sa}},\mathrm{lL,qL}}^{[3,\,\mathrm{fl1}]}}{256\pi^2 M_{\mathrm{Sa}}^2} + \cdots, \qquad (4.6)$$

where $\cdots$ corresponds to terms proportional to SM couplings. Both of these models seem interesting; we can take larger values for the quadratic couplings and obtain results that fit the $\chi^2$ in (4.2) better than the SM. However, this $\chi^2$ should be seen as only pointing us to a pattern; to accurately explore the low-energy phenomenology we will use the packages flavio [45] and smelli [45–47] which include a wide range of observables.

Extending the analysis to more observables, we quickly find an issue with the obtained patterns. For the $Q_5$ extension, we would need $\lambda_{\bar{\phi},\mathrm{dR,Fa}}^{[2]}\bar{\lambda}_{\bar{\phi},\mathrm{dR,Fa}}^{[3]} \sim 1$ to obtain a large enough $C_{\ell d}$; this is problematic because the same couplings generate $[C_{\phi d}]_{23}$ at tree-level, which would be excluded from measurements of $\mathcal{B}(B_s \to \mu\mu)$. On the other hand, for the $T_2$ extension, a large value for the quadratic couplings would be needed, $\lambda_{\mathrm{dR,Fa,Sa}}^{[2]}\bar{\lambda}_{\mathrm{dR,Fa,Sa}}^{[3]} \sim 5$ (allowed by perturbative unitarity). In turn, these couplings generate, at one-loop, $[C_{dd}]_{2323}$:

In[19]:=  `Limit[Match2Warsaw[`
          `alphaOdd[2, 3, 2, 3], {Sa -> {3, 3, -1/3},`
          `Fa -> {3, 3, 2/3}}], MFa -> MSa]/.onelooporder->1 // NiceOutput`

$$[C_{dd}]_{2323} = -\frac{\left(\lambda_{\overline{\mathrm{Fa}},\mathrm{dR,Sa}}^{[3]}\right)^2 \left(\bar{\lambda}_{\overline{\mathrm{Fa}},\mathrm{dR,Sa}}^{[2]}\right)^2}{64\pi^2 M_{\mathrm{Sa}}^2}, \qquad (4.7)$$

which would be excluded from $\overline{B_s} - B_s$ mixing. The remaining models including $S_3$ suffer from the same phenomenological constraints.

Let us now consider possible two-field extensions including, $\Pi_1 \sim (3, 2, 1/6)$ to generate $C_{\ell d}$ at tree-level (once again it is the only non-vector extension to do so [13]) and explore the possibility of generating $C_{\ell q}^{(1,3)}$ at loop-level. Following the same strategy as before, that is, finding the models in the output of `ListModelsWarsaw[alphaOlq1,True]` which also contain $\Pi_1 \sim (3, 2, 1/6)$ , we find several promising models at first. Once again however, they are clearly excluded due to the large couplings necessary that affect $b - s$ transitions.

Finally, to exhaust all two-field possibilities, we explore models which contribute only at loop-level to both $C_{\ell d}$ and $C_{\ell q}^{(1,3)}$ with

```
In[20]:= listmodelsOld=ListValidQNs[Select[ListModelsWarsaw[alphaOld[i, j, k, l]
        , True][[1, ;; ]],Length[#[[1]]]===2&], 8, 3];
        listmodelsOldinput=Flatten[Table[Table[(SOLDInputForm[#] & /@ ii2)
         , {ii2, ii1}], {ii1, listmodelsOld}],1];
        listmodelsOlq=ListValidQNs[Select[ListModelsWarsaw[alphaOlq3[i, j, k, l]
        , True][[1, ;; ]],Length[#[[1]]]===2&], 8, 3];
        listmodelsOlqinput=Flatten[Table[Table[(SOLDInputForm[#] & /@ ii2)
         , {ii2, ii1}], {ii1, listmodelsOlq}],1];
        modelintersection =
        Intersection[ listmodelsOldinput /.
        List[List[a_, b_], c_, d_] /; b > a :> List[List[b, a], c, -d]
        /. Sb -> Sa,  listmodelsOlqinput /.
          List[List[a_, b_], c_, d_] /; b > a :> List[List[b, a], c, -d] /.
         Sb -> Sa]
```

where in the last line we are taking the representations to a "standard" form in order to find the intersection between the models generating $C_{\ell q}^{(3)}$ and $C_{\ell d}$.

Among these models, the most promising (*i.e.* with the lowest numerical suppression) is $\Phi \sim (8, 2, 1/2) + \Psi \sim (8, 1, 0)$ with matching conditions for $C_{\ell q}^{(1,3)}$ given by:

```
In[21]:=    Limit[Match2Warsaw[
          alphaOlq1[3, 3, 3, 2], {Sa -> {8, 2, 1/2},
          Fa -> {8,1,0}}], MFa -> MSa]/.onelooporder->1 // NiceOutput
```

$$
\left[C_{\ell q}^{(1)}\right]_{3332} = \frac{\lambda_{\overline{\mathrm{Sa}},\mathrm{Fa},\mathrm{lL}}^{[3]}\bar{\lambda}_{\overline{\mathrm{Sa}},\mathrm{Fa},\mathrm{lL}}^{[3]}\lambda_{\overline{\mathrm{dR}},\mathrm{qL},\mathrm{Sa}}^{[\mathrm{fl1},\,2]}\bar{\lambda}_{\overline{\mathrm{dR}},\mathrm{qL},\mathrm{Sa}}^{[\mathrm{fl1},\,3]}}{192\pi^2 M_{\mathrm{Sa}}^2}
$$
$$
- \frac{\bar{\lambda}_{\overline{\mathrm{qL}},\mathrm{Sa},\mathrm{uR}}^{[2,\,\mathrm{fl1}]}\lambda_{\overline{\mathrm{Sa}},\mathrm{Fa},\mathrm{lL}}^{[3]}\bar{\lambda}_{\overline{\mathrm{Sa}},\mathrm{Fa},\mathrm{lL}}^{[3]}\lambda_{\overline{\mathrm{qL}},\mathrm{Sa},\mathrm{uR}}^{[3,\,\mathrm{fl1}]}}{192\pi^2 M_{\mathrm{Sa}}^2} \tag{4.8}
$$

and

```
In[22]:=    Limit[Match2Warsaw[
          alphaOlq3[3, 3, 3, 2], {Sa -> {8, 2, 1/2},
          Fa -> {8,1,0}}], MFa -> MSa]/.onelooporder->1 // NiceOutput
```

$$\left[C_{\ell q}^{(3)}\right]_{3332} = -\frac{\lambda_{\overline{\mathrm{Sa}},\mathrm{Fa},\mathrm{lL}}{}^{[3]}\,\bar\lambda_{\overline{\mathrm{Sa}},\mathrm{Fa},\mathrm{lL}}{}^{[3]}\,\lambda_{\overline{\mathrm{dR}},\mathrm{qL},\mathrm{Sa}}{}^{[\mathrm{fl1},\,2]}\,\bar\lambda_{\overline{\mathrm{dR}},\mathrm{qL},\mathrm{Sa}}{}^{[\mathrm{fl1},\,3]}}{192\pi^2 M_{\mathrm{Sa}}^2}$$
$$-\frac{\bar\lambda_{\overline{\mathrm{qL}},\mathrm{Sa},\mathrm{uR}}{}^{[2,\,\mathrm{fl1}]}\,\lambda_{\overline{\mathrm{Sa}},\mathrm{Fa},\mathrm{lL}}{}^{[3]}\,\bar\lambda_{\overline{\mathrm{Sa}},\mathrm{Fa},\mathrm{lL}}{}^{[3]}\,\lambda_{\overline{\mathrm{qL}},\mathrm{Sa},\mathrm{uR}}{}^{[3,\,\mathrm{fl1}]}}{192\pi^2 M_{\mathrm{Sa}}^2}\,. \tag{4.9}$$

Only the term proportional to $\lambda_{\overline{\mathrm{dR}},\mathrm{qL},\mathrm{Sa}}{}^{[\mathrm{fl1},\,2]}$ is responsible for the generation of $C_{\ell d}$ and so it results in $C_{\ell q}^{(1)} = -C_{\ell q}^{(3)}$.

## 4.3 Three-field extensions

Having exhausted all the two-field extensions and not finding a promising candidate to alleviate the tension while avoiding the exclusion from other observables, we will now explore models with three different heavy fields. Motivated by the same logic, we will focus on extensions containing either the $S_3$ or the $\Pi_1$ leptoquarks to have a mixed tree- and loop-level solution.

The list of models including $S_3$ but not $\Pi_1$ that can generate a loop-level $C_{\ell d}$ can be extracted in the following way:

```
In[23]:= listmodels=ListValidQNs[ListModelsWarsaw[alphaOld[i, j, k, l]
        , True][[ ;; ]], 3, 3];
        listmodelsinput=Flatten[Table[Table[(SOLDInputForm[#] & /@ ii2)
         , {ii2, ii1}], {ii1, listmodels}],1];
        Select[listmodelsinput, Length[#] === 3 &];
        Select[%,
        MatchQ[#, {___, Sa_ -> {{1, 0}, 3, hyp_}, ___} /;
          (StringContainsQ[ToString[Sa], "S"] && (hyp === -1/3
          || ! NumericQ[hyp]))]
          || MatchQ[#, {___, Sa_ -> {{0, 1}, 3, hyp_}, ___} /;
          (StringContainsQ[ToString[Sa], "S"] && (hyp === 1/3
          || ! NumericQ[hyp]))]&];
        Select[%,
          !MatchQ[#,( {___, Sa_ -> {{1, 0}, 2, 1/6}, ___} |
        {___, Sa_ -> {{0, 1}, 2, -1/6}, ___} )/;
            StringContainsQ[ToString[Sa], "S"]] &]
```

The three first instructions select the models with three fields (up to triplet representations), in SOLD input form, that can generate $C_{\ell d}$ without SM couplings. The next instruction extracts those models which contain $S_3$ (or its conjugate) either explicitly or with a generic hypercharge that could include $S_3$. Finally, we eliminate from the list those that contain $\Pi_1$.

Conversely, the list of models including $\Pi_1$ but not $S_1$ or $S_3$ that can generate a loop-level $C_{\ell q}^{(1,3)}$ can be obtained via:

```
In[24]:= listmodels=ListValidQNs[ListModelsWarsaw[alphaOlq3[i, j, k, l]
        , True][[ ;; ]], 3, 3];
```

```
listmodelsinput=Flatten[Table[Table[(SOLDInputForm[#] & /@ ii2)
 , {ii2, ii1}], {ii1, listmodels}],1];
Select[listmodelsinput, Length[#] === 3 &];
Select[%,
MatchQ[#, {___, Sa_ -> {{1, 0}, 2, hyp_}, ___} /;
  (StringContainsQ[ToString[Sa], "S"] && (hyp === 1/6
  || ! NumericQ[hyp]))]
  || MatchQ[#, {___, Sa_ -> {{0, 1}, 2, hyp_}, ___} /;
  (StringContainsQ[ToString[Sa], "S"] && (hyp === -1/6
  || ! NumericQ[hyp]))]&];
  modelswithPi1 = Select[%,
  !MatchQ[#, ({___, Sa_ -> {{1, 0}, 3, -1/3}, ___} |
  {___, Sa_ -> {{0, 1}, 3, 1/3}, ___} |
  {___, Sa_ -> {{0, 1}, 1, 1/3}, ___} |
  {___, Sa_ -> {{1, 0}, 1, -1/3}, ___}) /;
   StringContainsQ[ToString[Sa], "S"]] &]
```

Once again, generating a sizable loop effect in $C_{\ell d}$ (or alternatively $C_{\ell q}^{(1)}, C_{\ell q}^{(3)}$) requires $\lambda_{\mathrm{dR,F,S}}[2]\lambda_{\mathrm{dR,F,S}}[3] \sim \mathcal{O}(1)$ (or $\lambda_{\mathrm{qL,F,S}}[2]\lambda_{\mathrm{qL,F,S}}[3] \sim \mathcal{O}(1)$), which in turn can generate sizable contributions to the $O_{dd}, O_{qd}^{(1,8)}$ four-quark operators (alternatively to $O_{qq}^{(1,3)}, O_{qd}^{(1,8)}$). A successful completion has therefore to rely on a cancellation of the experimentally constrained contributions from the flavour off-diagonal entries in these operators. These tight experimental constraints arise mostly from the mass difference in the $B_s - \overline{B_s}$ system, or from the CP-violating part in the mixing of the $D^0 - \overline{D^0}$ system. The latter contribution is proportional to the SM CP-violation phase as we are not introducing new phases.

After examining the list of models obtained in `modelswithPi1`, we find that a cancellation of the $[2, 3, 2, 3]$ flavour entry in the $O_{qq}^{(1,3)}$ operators arises in the $\Pi_7 + N$ extension, where $N \sim (1, 1, 0)$. The cancellation of $\left[C_{qq}^{(1,3)}\right]_{2323}$ arises only when going to the Warsaw basis and it can be easily observed with:

In[25]:=
```
    Limit[Match2Warsaw[
    alphaOqq1[2, 3, 2, 3], {Sa -> {3, 2, 1/6},
     Fa -> {1,1,0}}], MFa -> MSa]/.onelooporder->1 // NiceOutput
```

In[26]:=
```
    Limit[Match2Warsaw[
    alphaOqq3[2, 3, 2, 3], {Sa -> {3, 2, 1/6},
     Fa -> {1,1,0}}], MFa -> MSa]/.onelooporder->1 // NiceOutput ,
```

which both output zero. At the Green's basis level this is not the case:

In[27]:=
```
    Limit[Match2Green[
    alphaOqq1[2, 3, 2, 3], {Sa -> {3, 2, 1/6},
     Fa -> {1,1,0}}], MFa -> MSa]/.onelooporder->1 // NiceOutput
```

$$\left[C_{qq}^{(1)}\right]_{2323} = -\frac{\left(\bar{\lambda}_{\overline{\text{Sa}},\text{Fa},\text{qL}}^{[2]}\right)^2 \left(\lambda_{\overline{\text{Sa}},\text{Fa},\text{qL}}^{[3]}\right)^2}{2304\pi^2 M_{\text{Sa}}^2} \tag{4.10}$$

This cancellation is reminiscent of the *magic* zero encountered for the dipole operator in Ref. [53] and further studied in [54, 55]. Let us note that the heavy field $N \sim (1,1,0)$ was also part of this cancellation. Complete dictionaries can be extremely useful in finding these apparently accidental cancellations more systematically, which can have important phenomenological consequences, allowing certain models to avoid tight experimental constraints.

To actually generate $C_{\ell q}^{(1,3)}$ we add the fermion $D \sim (3,1,2/3)$, resulting in:

In[28]:=   Limit[Match2Warsaw[
     alpha0lq1[3,3,3,2], {Sa -> {3, 2, 1/6},
     Fa -> {1,1,0}, Fb->{3,1,2/3}}], {MFa -> MSa,MFb->MFa}]
     /.onelooporder->1 // NiceOutput

$$\left[C_{lq}^{(1)}\right]_{3332} = -\frac{\lambda_{\overline{\text{Sa}},\text{Fa},\text{qL}}^{[2]}\,\bar{\lambda}_{\overline{\text{Sa}},\text{Fa},\text{qL}}^{[3]}\,\lambda_{\overline{\text{Sa}},\text{Fb},\text{lL}}^{[3]}\,\bar{\lambda}_{\overline{\text{Sa}},\text{Fb},\text{lL}}^{[3]}}{384\pi^2 M_{\text{Sa}}^2} + \cdots, \tag{4.11}$$

In[29]:=   Limit[Match2Warsaw[
     alpha0lq3[3,3,3,2], {Sa -> {3, 2, 1/6},
     Fa -> {1,1,0}, Fb->{3,1,2/3}}], {MFa -> MSa,MFb->MFa}]
     /.onelooporder->1 // NiceOutput

$$\left[C_{lq}^{(3)}\right]_{3332} = -\frac{\lambda_{\overline{\text{Sa}},\text{Fa},\text{qL}}^{[2]}\,\bar{\lambda}_{\overline{\text{Sa}},\text{Fa},\text{qL}}^{[3]}\,\lambda_{\overline{\text{Sa}},\text{Fb},\text{lL}}^{[3]}\,\bar{\lambda}_{\overline{\text{Sa}},\text{Fb},\text{lL}}^{[3]}}{384\pi^2 M_{\text{Sa}}^2} + \cdots, \tag{4.12}$$

where we have only included the contributions including the 3 heavy fields and non-linear couplings among them. Once again, the Lagrangian associated with this model can be obtained with CreateLag[ {Sa -> {3, 2, 1/6}, Fa -> {1,1,0}, Fb->{3,1,2/3}}].

The next step as performed in section 4.2 is to verify whether this model accommodates the pattern in (4.2) while still being in agreement with tight experimental constraints. We match this complete model and output it to smelli. We find that, despite the important cancellation for $\left[C_{qq}^{(1,8)}\right]_{2323}$, the model still generates important contributions to $B_s - \overline{B_s}$, be it through other flavor entries of $C_{qq}^{(1,8)}$ or through $C_{qd}^{(1,8)}$.

Due to this, we find that this model cannot saturate the large values of WCs implied by (4.2), but can accommodate the pattern. Furthermore, it avoids introducing any significant tension with regards to the SM prediction within the observables included in smelli. Indeed, after a brief scan over the parameter space, we find that the point

$$\lambda_{\overline{\text{Sa}},\text{Fa},\text{qL}}^{[2]} = 1, \qquad \bar{\lambda}_{\overline{\text{Sa}},\text{Fa},\text{qL}}^{[3]} = -2.2, \qquad \lambda_{\overline{\text{Sa}},\text{Fb},\text{lL}}^{[3]} = 4.1,$$
$$\lambda_{\overline{\text{dR}},\text{lL},\text{Sa}}^{[2,\,3]} = -0.05, \qquad \lambda_{\overline{\text{dR}},\text{lL},\text{Sa}}^{[2,\,3]} = 0.22, \tag{4.13}$$

considering the masses of the particles at 1 TeV, is in agreement with experimental constraints (with pull from the SM prediction always smaller than $\sim 2\sigma$) and results in a $\Delta\chi = \chi^{SM} - \chi^{UV} \approx 8$ with two degrees of freedom[6]. Note that the large value for the coupling $\lambda_{\overline{\mathrm{Sa}},\mathrm{Fb},\mathrm{lL}}{}^{[3]}$ was chosen to saturate the perturbative unitarity bound, $\lambda_{\overline{\mathrm{Sa}},\mathrm{Fb},\mathrm{lL}}{}^{[3]} \lesssim \sqrt{16\pi/3}$ [56–58]. There are also bounds from direct searches to be taken into account. For the $\Pi_1$ leptoquark in our setup, a 1 TeV particle is on the limit of what Ref. [59, 60] bounds; small modifications to the model could help avoid these constraints without affecting our general results.

Let us finally stress that we have taken several simplifying assumptions, in particular that all particles had the same mass, and that there was no flavor in the heavy sector. Although these assumptions do not alter our discussion and the usefulness of SOLD in phenomenological studies, it could in principle open more regions in the UV space of theories. At the same time, a more detailed study would require a fit to the parameters of the UV theory minimizing a global likelihood including the addressed tension along with the rest of observables considered, and correlations between them.

## 5    Conclusions and outlook

Effective Field Theories present an efficient setup to tackle the challenging search for new physics, following from the complementary use of the bottom-up and top-down approaches. However, an inefficient connection between them compromises the whole rationale. This can be avoided through the development of UV/IR dictionaries, that comprise the information of all models that can generate a set of operators and the matching conditions of those UV models.

In this work we presented the complete one-loop UV/IR dictionary for the SMEFT at dimension six for extensions with heavy fermions and scalars. The results are encoded in the latest version of the SOLD package. We have not only extended the previous dictionary to include all SMEFT operators, but we have also isolated the subset of the dictionary generated only by new physics couplings as an independent dictionary – this is particularly relevant when small SM couplings can be neglected. The package SOLD has also been upgraded to improve its performance and provide several new functions to facilitate its application to phenomenological studies. Furthermore, the tree-level dictionary is now also included in the package for convenience.

We show how SOLD is envisaged to readily go from the EFT to the UV and back, and allow the user to quickly obtain an idea of the phenomenology of a new physics model. We applied it to perform a systematic study of the possibilities to alleviate an observed tension in $\mathcal{B}(B \to K\overline{\nu}\nu)$ decays through one-loop UV solutions. Our results show that explaining the large deviation while being consistent with other experimental measurements in $b - s$ transitions is challenging. Through a systematic exploration, we found it impossible to explain this IR pattern at one-loop with two fields. Allowing for three-field extensions, a mixture of tree- and loop-level solutions was possible given an unexpected cancellation; this

---

[6]Despite using five different couplings, the $\chi^2$ from Ref. [44] only depends on two different combinations of them.

is reminiscent of the magic zero obtained by Ref. [53, 54] for the dipole operators. With this cancellation, it was possible to explain the IR pattern qualitatively, but difficult to obtain the large central values for the WCs that the current tension seems to imply, while avoiding other constraints. If the central value is updated to result in smaller WCs, loop-level solutions then provide compelling UV avenues to pursue in regards to this tension. The connection with a larger set of observables was possible by connecting the output of `MatchmakerEFT` with `smelli`.

While other tools like `MatchmakerEFT` or `Matchete` are more powerful to do a complete matching of a specific model (and therefore the most convenient choice for a detailed study of its complete phenomenology) `SOLD` partially trades efficiency with flexibility, allowing the quick extraction of information about the structure and size of contributions and patterns between coefficients in many different models. Nevertheless, it also facilitates the use of `Matchmakereft` for a posterior analysis. Moreover, it is the only tool that translates the SMEFT to the UV (as opposed to matching the UV onto the SMEFT), enabling an efficient connection between bottom-up and top-down approaches.

There are several directions to pursue in expanding this work and `SOLD`. The package itself will be under continuous improvement, both from a perspective of optimization, as well as to include further functions that the community suggests. Furthermore, including heavy vectors would also be extremely useful, given their relevance both from a model-building and phenomenological perspectives. In terms of `SOLD` usage, the goal is for it to become a pocket guide for model builders in the connection of the SMEFT with the UV. This can take two forms: to explain possible future tensions in data or explore the constraining power of future experiments.

## Acknowledgments

We are especially grateful to José Santiago for ongoing discussions and fundamental contributions in the earlier stages of this work. We also thank Lukas Allwicher and Maria Ramos for useful discussions. We thank Barbara Anna Erdelyi and Nudžeim Selimović for beta testing `SOLD`. The work of GG is supported by the Deutsche Forschungsgemeinschaft (DFG, German Research Foundation) under grant 491245950 and under Germany's Excellence Strategy — EXC 2121 "Quantum Universe" — 39083330. The work of PO is supported by the European Union's Horizon 2020 research and innovation programme under the Marie Sklodowska-Curie grant agreement n. 101086085 – ASYMMETRY by the INFN Iniziativa Specifica APINE, and by the Italian MUR Departments of Excellence grant 2023-2027 "Quantum Frontiers".

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
