# Peer review of "From the EFT to the UV: the complete SMEFT one-loop dictionary"

_SciPost Physics_

## Round 1 · Referee Report · Anonymous (Referee 1) · 2025-10-26

Strengths

This program is certainly an important addition to the EFT literature and relevant for future EFT analysis.

Weaknesses

The title and abstract of the manuscript are not well justified in the manuscript. I have pointed out my queries related to that. Once the authors reply to that, it will be clear what a user can compute using this program.

Report

The authors have claimed that this article is a complete SMEFT one-loop dictionary. They have presented this one-loop dictionary of dimension-six with an arbitrary number of scalars and fermions. I cannot recommend publishing this article unless all my queries are clarified by the authors.

Requested changes

It is not obvious that this program can be easily applied to compute the same beyond dimension six. Thus, the article's title is misleading. I suggest mentioning dimension-six in the title of the manuscript, e.g., “From the EFT to the UV: the complete dimension-six SMEFT one-loop dictionary”.

I have a few queries regarding the manuscript that I failed to note:

It is not clear whether this program can handle an arbitrary number of non-degenerate scalars and fermions.

Can this program be used to integrate out the scalar and fermions together?

How does this program take care of loops that consist of the light-heavy mixed propagators? How do they separate the local and non-local contributions?

Does this program address the emergence of dimension-six CP-violating operators?

Apart from these basic queries, I have a few generic comments:

  1. This manuscript seriously lacks proper referencing. They must cite all the efforts made so far to build the EFT program. I am naming a few here (if they are not cited already):

https://inspirehep.net/literature/1683160 https://inspirehep.net/literature/1389176 https://inspirehep.net/literature/1591722 https://inspirehep.net/literature/1631353 https://inspirehep.net/literature/1667740 https://inspirehep.net/literature/1332938 https://inspirehep.net/literature/1442364 https://inspirehep.net/literature/219220 https://inspirehep.net/literature/219217 https://inspirehep.net/literature/1374234 https://inspirehep.net/literature/1418809 https://inspirehep.net/literature/1474691 https://inspirehep.net/literature/1444897 https://inspirehep.net/literature/1835607 https://inspirehep.net/literature/1854479 https://inspirehep.net/literature/1852352

The title of the manuscript suggests that the EFT guides the choice of UV models. But I fail to have one. The authors must include a discussion of how EFT guides the choice of possible UV theories to justify the title of the manuscript, “From the EFT to the UV:….” In this context, the authors must add a discussion that relies on the following references:

https://inspirehep.net/literature/1852818 https://inspirehep.net/literature/2075735 https://inspirehep.net/literature/1898320 https://inspirehep.net/literature/2064830 https://inspirehep.net/literature/2127403

  1. Assignment of quantum numbers/representations for product groups in the function “ListModelsWarsawTree[alphaOlq1[i,j,k,l]]” is misleading when the group is SU(3) \otimes SU(2), then the prepresentation is depicted as (3 \otimes 1). Since the representations belong to different groups, the tensor product of two representations is not well-defined. That should be written as (3, 1), where the ordering of representations follows the same order as the group is defined. This should be consistent throughout the manuscript for other functions as well.

Recommendation

Ask for major revision

---

## Round 1 · Referee Report · Jaco ter Hoeve (Referee 2) · 2025-10-27

Strengths

  1. Presents the first tool that allows translating the space of Wilson coefficients to the space of UV models, rather than the other way around.
  2. Phenomenologically relevant: the authors demonstrate explicitly how SOLD helps efficiently map out certain classes of UV models that induce a certain pattern of known EFT coefficients in the IR. This makes it a useful tool for the wider community.
  3. The SOLD package is easy to install, user friendly and well documented and presented.

Weaknesses

  1. A few syntax errors and warnings are raised by Mathematica when running SOLD (see report and requested changes).
  2. The phenomenological section that uses smelli and flavio could be made clearer by adding a couple of supporting plots.

Report

The manuscript introduces a new release of the package SOLD that allows users to systematically map, for the first time, the space of EFT parameters to the space of UV models. In comparison to the previous release, the authors added support for the full set of of SMEFT operators and account for UV models that generate a given EFT parameter already at tree-level, as opposed to only at one-loop. The feature to go from the SMEFT back to the UV is genuinely new and provides a key contribution to the field. Therefore, I regard the contribution acceptable for publication, but a couple of suggestions and comments that should be addressed first.

Requested changes

  1. Unfortunately, I am not able to reproduce Figure 2 of the main manuscript. I am on Mathematica v14.3. I type ListOperators[{Sa -> {3, 1, -1/3}}, True] , which then gives me Figure 2 except for green ticks at tree level for operators OmuH2, Olambdad, Olambdae and Old. Everything else agrees. See attached file for more details (point 1)
  2. I have tried whether SOLD "closes": can I go back and forth between the SMEFT and the UV? For instance, ListModelsWarsaw[alphaOuG[i, j]] gives, among many other models, model {-3, 1, 1/3}, but when I do ListOperators[{Sa-> -3, 1, 1/3}] I do not retrieve OuG. Please check whether this is correct. ListOperators also gives me PreDrecrement errors, please see the attached file for more details (point 2).
  3. Please remind the reader of the notation Sa and Fa. I suspect these stand for Scalar and Fermion. It is also a bit unclear to me why all EFT operators are written with alpha in front, e.g. alphaOlq1. Why not just Olq1? Why this choice specifically?
  4. Just below Eq. 3.1, page 9: In[4] contains a typo: the replacement for Sa misses surrounding curly braces. Please correct.
  5. I spotted a minor grammatical mistake on page 9: this numbers distinguishes -> these numbers distinguish.
  6. Page 16, first paragraph: it says "pick models that include S3~(3, 3, -1/3)", but a bit later on underneath In[15] it says "selecting those that do not include S3 ~(3, 3, -1/3)". This seems contradictory, maybe a typo?
  7. Page 17, last paragraph: notation S3 + Q5 ~ (3, 2, -5/6). You probably mean just Q5 ~ (3, 2, -5/6), but it reads like the representation belongs to S3 and Q5 combined. Writing S3 + Q5 with Q5 ~ (3, 2, -5/6) should fix this.
  8. The part on page 17 that introduces Flavio and smelli could do with some plots to illustrate the tension with existing measurements. That way the arguments become easier to follow for the reader.
  9. What syntax should be used to list operators generated by a model defined in terms of a product of representations?
  10. Ref. 46 from the SMEFiT collaboration has been superseded by https://inspirehep.net/literature/2779255 and https://inspirehep.net/literature/2895783. Please update.

Attachment

Recommendation

Publish (easily meets expectations and criteria for this Journal; among top 50%)

---

## Round 1 · Referee Report · Anonymous (Referee 3) · 2025-11-8

Report

Warnings issued while processing user-supplied markup:

  • Inconsistency: plain/Markdown and reStructuredText syntaxes are mixed. Markdown will be used.
    Add "#coerce:reST" or "#coerce:plain" as the first line of your text to force reStructuredText or no markup.
    You may also contact the helpdesk if the formatting is incorrect and you are unable to edit your text.

In this work, the authors present an updated version of the code SOLD, which provides a dictionary between the SMEFT and all possible multi-particle SM extensions at one-loop level, containing only heavy scalars and fermions. In particular, the tool allows to determine the quantum numbers (or constraints on their combination) for all particles that are required in all n-particle one-loop models that generate this operator. In addition, the code also allows to determine the one-loop matching conditions onto the full SMEFT for any of the models thus obtained. This dictionary can certainly be useful for the model building community, as it allows to get a better overview of the BSM model landscape required for specific IR scenarios. The article is well written and mostly clear. However, I have several (mostly minor) questions: 1) Global symmetries can significantly modify the IR description of a theory. How would one treat such global symmetries in SOLD? 2) I suppose that SOLD is fully specific to the Warsaw basis. Is that the case, or is it possible to include also other bases? 3) Since the matching of the generic UV Lagrangian onto SMEFT is precomputed in SOLD, I am wondering to which order in the number of BSM fields this is done? Or in other words, how many BSM fields are possible to include when running the code? 4) It is not quite clear to me in which cases one needs to use MatchMakerEFT in order to obtain the full one-loop matching of a given model, or when this can be done with SOLD. A clarification for this would be good. 5) It would be useful to state the $\gamma_5$ scheme employed in the computation, given that subsequent calculations in the EFT are required to be performed in the same scheme. 6) The authors state that UV theories with a tree-level mass mixing are not supported. What is the specific reason behind that? As long as the mixing parameter is an IR scale, the matching should be still possible. 7) In Mathematica input In[4] it seems like there are some curly brackets missing. 8) Regarding figure 2: Why is the operator $O_{lequ}^{(3)}$ missing in the table, which is generated by the $S_1$? Also, assuming that the operators $O_{lambdau}$ and $O_{lambdad}$ represent the SMEFT up- and down-quark Yukawas, I would assume that the $S_1$ generates an up-Yukawa correction at one loop and not a down-Yukawa correction, contrary to what figure 2 claims.

While the example of $B \to K \nu \nu$ used in Sec. 4 to showcase the workings of SOLD might be not entirely ideal, due to the large NP effect that is required to satisfy the central value, making a one-loop explanation less compelling, I still think the example nicely highlights all benefits of using a dictionary like SOLD. However, the description is partially difficult to follow: 9) The code presented in Mathematica input In[15,20,23,24] is extremely difficult to read and a non-expert in Mathematica syntax will likely not be able to understand this. In my opinion it would be better to either move this code to an appendix or maybe better to create an example notebook as ancillary material, where one can directly evaluate the example. That way, the main part of the article could still be understood without substantial experience with Mathematica. 10) It appears that for inputs In[25] and In[26] the output is missing. 11) It might be good to add the BSM masses to the list of parameters in Eq. (4.13). I recommend a minor revision for this article and believe that once these minor issues and questions have been addressed, it will be suitable for publication in SciPost Physics.

Recommendation

Ask for minor revision

---

## Editorial Decision

awaiting_resubmission